# Inflammation: The Link between Neural and Vascular Impairment in the Diabetic Retina and Therapeutic Implications

**DOI:** 10.3390/ijms24108796

**Published:** 2023-05-15

**Authors:** Hugo Ramos, Cristina Hernández, Rafael Simó, Olga Simó-Servat

**Affiliations:** 1Diabetes and Metabolism Research Unit, Vall d’Hebron Research Institute, Universitat Autònoma de Barcelona, 08035 Barcelona, Spain; cristina.hernandez@vhir.org (C.H.);; 2Centro de Investigación Biomédica en Red de Diabetes y Enfermedades Metabólicas Asociadas (CIBERDEM), Instituto de Salud Carlos III (ICSIII), 28029 Madrid, Spain

**Keywords:** diabetes, diabetic retinopathy, retina, neurovascular unit, inflammation, cytokine

## Abstract

The etiology of diabetic retinopathy (DR) is complex, multifactorial and compromises all the elements of the retinal neurovascular unit (NVU). This diabetic complication has a chronic low-grade inflammatory component involving multiple inflammatory mediators and adhesion molecules. The diabetic milieu promotes reactive gliosis, pro-inflammatory cytokine production and leukocyte recruitment, which contribute to the disruption of the blood retinal barrier. The understanding and the continuous research of the mechanisms behind the strong inflammatory component of the disease allows the design of new therapeutic strategies to address this unmet medical need. In this context, the aim of this review article is to recapitulate the latest research on the role of inflammation in DR and to discuss the efficacy of currently administered anti-inflammatory treatments and those still under development.

## 1. Introduction

Diabetes mellitus (DM) is a chronic disease that embraces multiple metabolic disorders that share a common phenotype, an uncontrolled increase in blood glucose levels [1]. In accordance to the International Diabetes Federation, 415 million adults had DM in 2015, and this is expected to rise to 642 million by 2040, exacerbating the impact of this public health burden [2]. DM leads to a state of chronic hyperglycaemia that can damage several organ systems, triggering life-threatening and disabling health complications [2]. Chronic microvascular complications are diabetic retinopathy (DR), neuropathy and nephropathy while macrovascular complications include coronary heart disease, peripheral vascular disease and stroke [3]. These DM complications are responsible for much of the burden linked with diabetes [4].

DR is one of the most frequent complications of diabetes, affecting one-third of diabetic patients and being the leading cause of preventable vision loss among working-age adults [5,6]. Current treatments only address the advanced stages of the disease and are associated with severe side effects, giving rise to an unmet medical necessity [7]. In the context of finding an effective treatment against early stages, it is mandatory to mention the concept of the retinal neurovascular unit (NVU), a functional and interdependent coupling between the different retinal cells (neurons, glial cells, microglia, endothelial cells and pericytes) that integrates vascular flow with metabolic activity [8]. The NVU is early impaired during the pathogenesis of DR, and the inflammatory process that occurs in the retina plays a key role [8,9]. This review article is intended to recapitulate the latest research regarding the role of inflammation in DR and to discuss the effectiveness of the current and potential anti-inflammatory treatments.

## 2. Inflammation and Diabetic Retinopathy

Inflammation is a non-specific and protective response of the body against harmful agents [10]. The immune system is principally responsible for recognizing and attempting to eliminate these noxious stimuli, which include pathogens, damaged cells or irritants. In general terms, the acute inflammatory process involves several and different immune cells that release inflammatory mediators that promote vasodilation, which consequently increases blood flow and allows more blood and more immune cells (leukocytes, mainly neutrophils) to reach the injured tissue [11]. Inflammatory mediators also facilitate the passage of immune system cells out of small blood vessels, while activating the pain pathway by irritating nervous tissue in the affected area in order to alert the brain [12]. If the acute inflammation is sustained over time it can become chronic but most of the features are also maintained, the main difference being the replacement of leukocytes by macrophages and lymphocytes [13]. Therefore, the hallmarks of chronic inflammation are the infiltration of lymphocytes, macrophages and plasma cells, and the release of inflammatory cytokines, growth factors, and enzymes, thereby promoting the secondary repair of the injured tissue, which includes fibrosis and granuloma formation [14].

DR has a multifactorial and complex etiology where all the components of the retinal NVU are affected even before any clinical sign can be detected. In terms of inflammation, DR is considered a chronic low-grade inflammatory condition that involves multiple inflammatory mediators and adhesion molecules, with chronic hyperglycaemia being the main trigger [15,16]. The increase in the intracellular glucose levels results in the activation of four different cellular glucose metabolic pathways: polyol and hexosamine pathways, activation of protein kinase C (PKC) and increased advanced glycation end product (AGE) formation. All these processes, in one way or another, lead to a state of oxidative stress, inflammation and neurotrophic deficiencies, which consequently damage the NVU in terms of neuronal apoptosis, glial activation and microvascular abnormalities [17]. All these features are interconnected and upregulate inflammatory mediators, while triggering the recruitment and infiltration of neutrophils, macrophages and monocytes, resulting in an abnormal inflammatory response [15]. This inflammatory component of the disease is intimately associated with all the hallmarks of DR, contributing to the neurodegenerative process and the vascular abnormalities of the diabetic retina [18].

### 2.1. Cells Involved in the Inflammatory Response That Occurs in the Diabetic Retina

The NVU in the retina is composed of diverse neural cell types (i.e., ganglion cells, amacrine cells, horizontal cells, bipolar cells and photoreceptor cells), glia (Müller cells and astrocytes), professional immune cells (microglia and perivascular macrophages) and vascular cells (endothelial cells and pericytes) [19,20,21,22]. All of these components are in intimate communication and maintain the integrity of the inner blood retinal barrier (iBRB). The impairment of the neurovascular unit and the subsequent disruption of the iBRB is a primary event in the pathogenesis of DR [23,24,25,26], and it is summarized in Figure 1.

Among the first cells damaged during the early stages of DR, Müller cell injury plays a crucial role, especially with regard to pro-inflammatory pathways. Müller cells represent 90% of retinal glia and their geographical distribution throughout the retina establishes them as a key regulator of both physiological and pathological responses between neurons and vascular cells [27]. In addition, they are considered the main nutritional support of the retina because of its relatively higher rate of glycolysis [28]. Under the hyperglycaemic condition of DR, Müller cells can undergo early reactive gliosis, which is revealed by the aberrant expression of glial fibrillary acidic protein (GFAP). Gliosis implies morphological, biochemical, and physiological changes and it is associated with an overexpression of VEGF and innate immune-related pathways, resulting in an overexpression of proinflammatory cytokines and BRB dysfunction [29,30]. More precisely, Müller cell-derived VEGF contributes to the loss of occludin and zonula occludens-1 (tight junctions proteins) in endothelial cells, which are necessary for the integrity of the BRB, while it activates Nuclear factor-kappa-B (NF-κB) and causes the production of multiple pro-inflammatory cytokines such as intercellular adhesion molecule-1 (ICAM-1) or tumor necrosis factor-α (TNF-α) [27,31]. Furthermore, Müller cells create this chronic inflammatory environment that promotes retinal fibrosis during the proliferative stages of the disease [27].

With regard to the another macroglial cell, astrocytes, their distribution is linked to the presence of retinal blood vessels, and in physiological conditions they regulate ion homeostasis, neuronal signalling and the proper functionality of endothelial cells to preserve in the inner BRB properties [28]. During DR, the exposure of high-glucose levels alter the homeostasis of retinal astrocytes, increasing the production of inflammatory cytokines [mainly TNF-α and interleukin-1 beta (Il-1β)] and reactive oxygen species (ROS), which consequently impairs their proliferative, migrative and adhesive capacities [32]. Overall, macroglia exacerbate the neuroinflammatory environment which contributes to both neurodegeneration and microvascular abnormalities.

In addition to macrogial cells, activated microglia can also mediate diabetes-induced subclinical inflammation. Its activation has been postulated as an early event that could be triggered by the AGE/AGE receptor pathway [33,34], which can be detected even before the activation of macroglial cells [18,35]. Microglial activation is accompanied by a phenotype change toward an amoeboid shape and presents two opposite roles depending on the polarization of resident immune cells of the retina triggering proinflammatory (M1) or anti-inflammatory (M2) actions [30,31]. In early stages of DR, the M2 response occurs concurrently with the M1 response ameliorating inflammation and delaying the progression of the disease. However, during the progression of DR, the M1 response is maintained whereas the M2 response declines and the classical pro-inflammatory signaling pathways are chronically activated [31]. Their activation via NF-κβ and extracellular signal-regulated kinase (ERK) signaling pathways results in the release of various pro-inflammatory cytokines [mainly TNF-α, Il-1β and interleukin-6 (IL-6)], chemokines, caspases and glutamate [36]. These molecular mediators contribute to macroglial activation, to the disruption of the BRB and NVU impairment and to neuronal death. Cytokines are not only secreted by activated or reactive glial cells, but also by retinal pigment epithelium (RPE), photoreceptors and immune cells [37,38].

Blood circulating leukocytes play an important role in the endothelial damage that occurs in DR. Leucocytes engage with adhesion molecules such as ICAM-1, vascular cell adhesion protein-1 (VCAM-1) and selectins on the surface of the endothelial wall (leukostasis) provoking the occlusion of the capillaries. Such vascular immune-cell interactions also contribute to microvascular damage by releasing cytokines and superoxide via the respiratory burst, which alters the integrity of the NVU [37,39]. In advanced stages of DR where immune privilege is compromised, circulating immune cells and serum proteins may infiltrate the retina and the vitreous, thus participating in chronic inflammation which results in vascular and neuronal damage [39,40,41,42].

### 2.2. Inflammatory Cytokines and Their Role in DR Development

Several proinflammatory cytokines, chemokines and adhesion molecules have been found elevated within the vitreous of diabetic patients with advanced stages of DR [43,44]. The pro-inflammatory cytokines are generated as a response to oxidative stress and the activation of NF-κβ, resulting in a low-grade inflammatory state.

IL-1β is one of the main cytokines involved in diabetic complications. It is able to activate NF-κβ and induce the expression of adhesion molecules and chemoattractant factors by the endothelium. NF-κβ can be also activated by the receptor for advanced glycation end-product (RAGE) interaction with high mobility group box 1 protein (HMGB1), which also functions as a pro-inflammatory cytokine [45].

The activation of NF-κβ induces the expression of several pro-inflammatory cytokines. TNF-α is one of the most important and has been implicated in diabetes complications as it is able to amplify the inflammatory cascade. It should be mentioned that both IL-1β and TNF-α are able to increase vascular permeability and to induce the expression of adhesion molecules by endothelial cells [46,47,48]. Both interleukin-8 (IL-8) and IL-6 have been found elevated in the vitreous of patients with proliferative DR (PDR) [46]. IL-6 is pleiotropic cytokine which is able to increase endothelial cell permeability in vitro by rearranging actin filaments and by changing the shape of endothelial cells, while it promotes leukocyte recruitment by its interaction with endothelial cells [49,50]. IL-8 has been recognized as a potent chemoattractant and as an activator of neutrophils and T lymphocytes [51]. Monocyte chemoattractant protein-1 (MCP-1) is one of the main cytokines and it has been found in the vitreous from patients with PDR in the same range as that reported in pleural effusions of patients with pneumonia or tuberculosis, and its levels correlate with PDR activity [44]. Under hyperglycemic conditions MCP-1 is produced by endothelial cells, RPE cells and Müller glial cells [52]. All these cytokines increase the expression of adhesion molecules. The main adhesion molecule is ICAM-1 and an elevation of its soluble form has been found in DR [46]. Adhesion molecules such as ICAM-1 or VCAM-1 facilitate leukostasis, which is the irreversible adhesion of circulating leukocytes to the surface of endothelial cells. This phenomenon induces endothelial cells apoptosis, thus favoring vascular leakage [48]. Finally, stromal cell-derived factor I (SDF-1) is upregulated in damaged tissues and mobilizes stem/progenitor cells to promote their repair. SDF-1 acts through its receptor, the C-X-C chemokine receptor type 4 (CXCR4), at several key steps in the process of ischemic repair, such as the recruitment of endothelial progenitor cells (EPCs) from the bone marrow [53]. Moreover, SDF-1 induces VEGF expression [54]. The severity of SDF-1 effects has been found to be increased in diabetic patients who present the homozygous SDF-1 30′A genotype, which has been associated with higher insulin-dependent horiocapill of adult progenitor cells, with increased levels of SDF-1 mRNA in peripheral blood mononuclear cells and with a higher predisposition to suffer from PDR [55].

Proteomic analysis in the vitreous of the retinas from diabetic patients has contributed to the identification of the additional inflammatory mediators involved in DR. In this regard, two studies of the vitreous of diabetic subjects with PDR showed that several complement components were increased compared with control subjects [56,57]. In addition, inflammation-associated proteins such as alpha1-antitrypsin, apolipoproteinA4, albumin and transferrin were found to be significantly elevated in the vitreous of PDR patients [58]. In addition, two proteins related to inflammation were found differently expressed in the vitreous fluid of patients with diabetic macular oedema (DME) in comparison with PDR and non-diabetic subjects: hemopexin (increased), which is an acute phase reactant that increases permeability of BRB; and clusterin (decreased), which limits the inflammatory response after injury [46,59,60].

Gao et al. [61] demonstrated that both extracellular carbonic anhydrase-I and kallikrein-mediated innate inflammation were involved in the pathogenesis of DME. Plasma kallikrein is a member of the kallikrein-kinin system (KKS) and catalyzes the release of the bioactive peptide bradykinin, which induces inflammation, vasodilation and vessel permeability [62,63]. Preclinical experiments on diabetic animals showed that inhibition of KKS components was effective in decreasing retinal vascular permeability, and KKS is an emerging therapeutic target for the treatment of DME [64,65].

Complement activation is an important regulator and effector of the inflammatory process, and increased levels of several main components have been found in the vitreous fluid of diabetic patients with PDR [56,57]. Moreover, deposits of the membrane attack complex (MAC), which is the final effector of the complement cascade, have been reported within retinal blood vessels and choriocapillaries of human donors with diabetes [66,67], pointing to the role of the complement system in the vascular pathology of DR. The initial trigger of complement activation in DR could be one of multiple. First, complement factor H (CFH) regulates the complement system as a cofactor of complement factor I by inactivating C3b. CFH also accelerates the decay of the C3 convertase C3bBb of the alternative pathway, and competes with factor B for binding to C3B [68,69]. Lundh von Leithner et al. [69], using CFH-knockout mice demonstrated that the absence of CFH promoted the accumulation of C3 on the vascular endothelium in the neuroretina and the RPE-choroid interface, and it was associated with a reduced perfusion and leukostasis. In the same model, glial activation and thinning of the photoreceptor layer with a global increase in retinal thickness was observed [70]. Moreover, Wang et al. [71] investigated the association between CFH and factor B (CFB) gene polymorphisms and DR. CFH inhibits the alternative pathway of the complement, and CFB activates it. The genetic study revealed a significant increase in the frequencies of the A allele and AA genotype for rs1048709 (CFB in patients with DR compared to controls with diabetes. On the other hand, there was a significant decrease in the frequencies of the A allele and AA genotype for rs800292 (CFH) in patients with DR compared to diabetic controls. In addition, the study found that the rs800292/AA genotype was related with a delayed progression of DR. Therefore, the decrease in CFH and activation of CFB and the alternative pathway are involved in the pathogenesis of DR.

CD59 is a complement inhibitor of the final step in the MAC assembly. In an animal model of diabetes, human soluble CD59 (sCD59) transgene expressed from adeno-associated virus administered by an intravitreal injection attenuated the retinal vascular leakage, the non-perfusion areas and ganglion cells apoptosis. Interestingly, Zhang et al. [66], found a significant reduction in CD59 in retinal samples from diabetic donors, thus suggesting that the loss of this regulatory mechanism may be the cause of the increased complement activation that occur in DR. Finally, as mentioned above, clusterin is decreased in the vitreous fluid of patients with DME [72]. Clusterin inhibits the inflammatory response by binding C5B-9 complex, resulting in its inactivation [73]. Moreover, clusterin is also capable of inhibiting VEGF-induced hyperpermeability, thus abrogating BRB breakdown [74].

The aforementioned studies focused on advanced stages of DR but inflammation is also an important player in early DR [75]. In fact, the activation of glial cells represents one of the first signs of inflammation in DR and glial cells are one of the main sources of proinflammatory cytokines/chemokines [48,75]. In addition, it has been demonstrated that hyperglycemia in retinal Müller cells triggers the expression of acute-phase response proteins and other inflammation-related genes [75]. Therefore, glial activation is a primary event in the pathogenesis of DR that contributes both to neurodegeneration and early microvascular impairment [76]. Vujosevic et al. [42] performed a proteome analysis of the aqueous humour of diabetic subjects without DR in the fundus exam, with mild DR and controls, identifying several inflammatory cytokines that were differentially expressed among groups, including interferon gamma (INF-γ), IL-1β, interleukin-3 (IL-3), interleukin-10 (IL-10), interferon gamma-induced protein 10 (IP-10) and monocyte chemoattractant protein 2 (MCP-2). Moreover, in a proteomic analysis performed in early stages of diabetic retinopathy, we observed the activation of inflammation together with neurodegeneration related pathways [77].

## 3. Targeting Inflammation in DR

As has been discussed previously, pro-inflammatory mediators, such as TNF-α, IL-6, IL-1β or VEGF, play a key role in the early and advanced stages of DR contributing to all the hallmarks of the disease. Therefore, therapies that counteract the inflammatory process of the diabetic retina should be considered as a potential option for DR therapy [78]. Current and experimental treatments based on that principle have been conceived to stop inflammation by following different strategies. In this section, the most relevant approaches targeting DR-related inflammation will be reviewed.

### 3.1. Anti-Inflammatory Treatments

#### 3.1.1. Corticosteroids

Corticosteroids (also referred to as glucocorticosteroids, glucocorticoids or just steroids) are the most common anti-inflammatory drugs whose principal effect is to switch off multiple inflammatory genes that encode for cytokines, chemokines, inflammatory enzymes, adhesion molecules, receptors and proteins that have been activated during a chronic inflammatory process (genomic effect). When used at higher concentrations, they promote the synthesis of anti-inflammatory proteins and present non-genomic effects. [79]. Corticosteroids act through glucocorticoid receptors (GR), which are located in the cytoplasm and whose nuclear translocation is suppressed by chaperones. When corticosteroids bind to their receptors, the complex dissociates from the chaperones and translocates to the nucleus, where as a homodimer it activates the glucocorticoid response elements (GRE) of the DNA and regulates gene transcription, directly or indirectly affecting the production of more than 600 proteins involved in the inflammatory pathway [80]. The consequences of these complex anti-inflammatory mechanisms of action of have been summarized as the five Rs (ready, reinforce, repress, resolve, and restore). In summary, corticosteroids ready the innate immune system to respond to microbial products and tissue injury, increase the levels of circulating bone marrow-derived neutrophils, reinforce the immune system, promote leukocyte distribution, repress proinflammatory transcription factors and enhance anti-inflammatory cytokine production. Additionally, corticosteroids resolve inflammation by stimulating the secretion of Annexin-1, shifting T cell signaling toward a T helper 2 response, inducing neutrophil and T cell apoptosis, thus restoring wound healing and anti-inflammatory phenotype in macrophages [81]. The ability of this family of drugs to accomplish these pleiotropic actions will depend on several factors including post-translational modifications, extracellular environment, ligand availability, duration of signaling, cell type-specific cofactors, binding partners or chromatin accessibility [82].

The use of corticosteroids to treat inflammatory eye diseases can be traced back to the 1950s, and it was not until the 1990s that they were administered to test their effectiveness against DR and DME [83]. In that context, oral steroids were associated with multiple adverse effects at a systemic level, including diabetes exacerbation. However, intravitreal injections exerted beneficial properties in preclinical and clinical studies, becoming the preferred route of administration [84]. Despite this, intraocular administrations are not exempt from side effects, such as cataract formation, elevation of IOP and glaucoma, which limits their application. The Diabetic Retinopathy Clinical Research Network (DRCR.net) found that intravitreal corticosteroids could increase visual acuity and reduce DME in diabetic patients during the first 6 months after treatment, but not in the following 6 months where vision loss and cataract formation occurred [85]. The main beneficial effects of corticosteroids on the eye include anti-inflammatory properties, inhibition of leukocyte adhesion to vascular walls and tight junction preservation, inhibition of VEGF gene activation and anti-angiogenic capacity, and a reduction in vascular permeability, resulting in the reestablishment of the BRB breakdown [86]. Corticosteroids can also reduce DME through their non-genomic effects on the cytoplasmic membrane of vessels (reduction in permeability and vasoconstriction) and Müller cells (induction of adenosine production and diminishment of cellular swelling), which allow the reabsorption of the excess fluid in the extracellular space. Furthermore, recent data suggest that glucocorticoids have a neuroprotective effect as a consequence of all the aforementioned effects, which are able to arrest retinal thinning [87]. Nevertheless, all these effects could be transitory and frequently new injections are needed, with the risk that these entail [80,88].

Going deeper into the anti-inflammatory effects of corticosteroids (Figure 2), these are mainly due to the downregulation of the transcription factors NF-κB and Activator protein-1, which control the production of multiple proinflammatory cytokines, and to the inhibition of the enzyme phospholipase A2, whose levels are increased in diabetic retinopathy contributing to the production of ICAM-1, TNF-α and VEGF [86]. In addition, corticosteroids promote the secretion of anti-inflammatory proteins such as IL-10, adenosine, nuclear factor of kappa light polypeptide gene enhancer in B-cells inhibitor alpha (IκBα) and Annexin-1, the latter being essential in promoting neutrophil apoptosis at the site of inflammation. Glucocorticoesteroids also inhibit the adhesion and extravasation of neutrophils and induce tissue resident macrophages (MΦ) to undergo a phenotypic change to become M2-like or anti-inflammatory. These macrophages no longer produce proinflammatory cytokines. Instead, they produce IL-10, have enhanced phagocytic activity to remove apoptotic cells, and promote tissue healing. Finally, glucocorticoids also favor T cell apoptosis [81].

Intravitreal injections of steroids are mainly composed of triamcinolone acetonide (TA), fluocinolone acetonide (FA), or dexamethasone sodium phosphate (DEX), three different synthetic glucocorticoids without mineralocorticoid activity and with distinct advantages and disadvantages among them. They present different GR binding affinities and lipophilicities which explain their relatively variable effects [89]. Microarray studies in relevant ocular cells revealed that DEX, FA and TA present distinct expression patterns as well as unique biologic responses, which are dose and time dependent [90].

TA is administered as an injectable suspension composed of water-insoluble steroid crystals that allow for a longer duration of action (6 months) compared to other steroids [78]. Kenalog-40 is the most widely used TA (Bristol-Meyers Squibb, New York, NY, USA) even the intravitreal route is off-label [91,92]. On the other hand, DEX, which was the first corticosteroid intravitreally injected to treat DR, presents less side effects than TA but a higher water solubility, remaining in the vitreous for only a few weeks. Nevertheless, DEX is currently administered via a biodegradable implant (Ozurdex^®^, Allergan Inc., Irvine, CA, USA) that is inserted in the vitreous and releases DEX slowly [92]. The Ozurdex implant uses the NOVADURTM delivery system, which is formed by poly(D, L-lactide-co-glycolide) matrix microspheres that are degraded slowly in the vitreous and that gradually release 0.7 mg of DEX, increasing its effects to over 6 months [79,93]. Finally, FA is also provided in the form of an intravitreal implant (Iluvien^®^, Alimera Sciences, Alpharetta, GA, USA), in this case a non-degradable one, that releases 0.2µg/day of FA into the aqueous and vitreous humors during 3 years in a sustained manner over time [94]. The lower lipophilicity of FA in comparison to TA and DEX explains the longer durability while reducing side effects [79,93]. However, since all three glucocorticoids are associated with an increased risk of cataract progression, glaucoma and increased IOP, they are generally considered second-line agents for patients who do not respond adequately to anti-VEGF injections, which are the current first-line agents [89,95]. In addition, the usefulness of corticosteroids by topical route for treating DME in humans have been reported [96], which would avoid the adverse effects caused by the intravitreal route while ensuring the protective effects, although further research on this issue is needed.

#### 3.1.2. Nonsteroidal Anti-Inflammatory Drugs

Non-steroidal anti-inflammatory drugs (NSAIDs) are one of the most widely prescribed families of medicines and are routinely administered for their antipyretic, analgesic, and anti-inflammatory effects. NSAIDs are potent inhibitors of cyclooxygenase enzymes (COX) and consequently of the synthesis of proinflammatory prostaglandins (PGs), which contribute to the disruption of the BRB, induce vasodilatation and promote leukocyte migration in DR and DME. COX-2 is the predominant isoform in the RPE. All the aforementioned reasons suggest that NSAIDs could be beneficial for the inflammatory process that occurs in the retina during DR, and evidence exists of their efficacy in delaying disease progression in animal models [97]. In fact, topical administration of nepafenac, a COX-2 inhibitor, reduced DR abnormalities without side effects in streptozotocin-induced diabetic rats [98]. Nevertheless, the few available studies at clinical level do not show any clear benefit beyond a slight reduction in the development of microaneurysms during the early stages of the disease [99]. Therefore, it can be concluded that there are not sufficient clinical data to recommend NSAIDs for the treatment of these ocular diabetic complications [97]. However, the topical use of NSDAIDs has been shown to be effective in the prophylaxis and treatment of macular oedema caused by cataract surgery in patients with and without DR [100]. Due to the higher incidence of this type of complications in diabetic patients, NSAIDs are mostly used in this therapeutic context [101,102,103].

#### 3.1.3. TNF-α Blockade

Regarding drugs whose mechanism of action is based on the blockade of a single inflammatory molecule, those targeting TNF-α were considered potential candidates for the treatment of DR due to the key role that this molecule plays in its development. Briefly, TNF-α activates the NF-κB pathway, induces iNOS expression and ROS formation while it mediates leukostasis by upregulating adhesion molecules such as VCAM-1, ICAM-1 and E-selectin [84]. In addition, TNF-α has been associated with cell death of endothelial cells and retinal neurons in DR, contributing to BRB breakdown [99]. The effect of TNF-α blockers has already been explored in experimental models of DR. CNTO5048 and etanercept, two different TNF-α blockers, prevented inflammation, retinal leukostasis, apoptosis, formation of acellular capillary and BRB breakdown when administered subcutaneously in diabetic rats [104,105,106]. Sfikakis et al. demonstrated in a pilot clinical study with four patients that intravenous infliximab (a TNF-α blocking antibody) was able to improve visual acuity [107]. However, in another clinical study involving 39 eyes with refractory DR, intravitreal injections of infliximab and adalimumab (another TNF-α blocker) did not show any beneficial effect [108]. Although more research is needed to clarify the effectiveness of TNF-α blockade in DR, this therapeutic strategy may be considered as a promising alternative in those patients who do not respond to classical glucocorticoid therapies or in whom chronic glucocorticoid therapy causes undesirable side effects [109].

#### 3.1.4. Blocking Other Specific Inflammatory Mediators

Apart from TNF-α, there are other inflammatory molecules with an important enough role to consider specific blockades against them, such as IL-1β or the integrin leukocyte function associated antigen-1 (LFA-1) [80]. Systemic canakinumab, a selective antibody against IL-1β, halted the progression of neovascularization but no sign of regression was observed [110]. Regarding LFA-1, an integrin expressed in leukocytes that mediates leukostasis, its inhibition by topical administration of the SAR 1118 antagonist prevented vascular leakage and leukostasis in diabetic rats [111]. Nonetheless, further studies are needed.

#### 3.1.5. Suppressors of Cytokine Signaling (SOCS) Proteins

Suppressors of cytokine signaling (SOCS) comprise a family of proteins that regulate the initiation, intensity, duration and quality of cytokine responses through a classical negative feedback loop. SOCS-1, whose expression increases during ocular inflammatory processes, modulates the functions of M1/M2 macrophages and negatively regulates IL-6 and TNF-α. In experimental DR, topical administration of the peptidomimetic SOCS1 reduced microglial and macroglial activation and proinflammatory cytokines, in turn reducing apoptosis, glutamate excitotoxicity and functional abnormalities in the diabetic retina [112]. Nevertheless, specific clinical trials are required to assess their safety and efficacy

### 3.2. Antioxidants

Oxidative stress is one of the main factors responsible for the onset and progression of the inflammatory process that takes place in the diabetic retina, as well as contributing to many of the other pathological processes such as mitochondrial damage, apoptosis in different cell types, abnormal microvascular changes or lipid peroxidation [113]. Therefore, antioxidants represent a promising therapeutic option as they are able to reduce inflammation at a much earlier stage while preventing the damage caused directly by oxidizing agents. As reviewed by Garcia-Medina et al., there are several experimental studies showing that that antioxidants are able to prevent and/or reverse experimental RD. The results in clinical trials are more controversial; however, some of them are quite promising, especially when antioxidant combinations are taken into consideration [114].

### 3.3. Blockers of the Renin–Angiotensin System

The systemic and local Renin-Angiotensin systems (RAS) are involved in the pathogenesis of DR, playing an important role in both diabetes and hypertension-induced retinal inflammation. In fact, several components of the RAS system are overexpressed in the retinas of diabetic patients, such as renin, angiotensin converting enzyme (ACE), angiotensin II or angiotensin II type 1 receptor (AT1R) [115]. Specific blockers of ACE (Enalapril, Captopril) and AT1R (Losartan, Candesartan) were effective in preventing inflammation, oxidative stress and vascular damage in the retinas of diabetic animal models [89,116]. Furthermore, a meta-analysis study including 21 randomized clinical trials concluded that systemic administration of RAS inhibitors in normotensive patients, and not in hypertensive patients, is associated with a lower risk of DR progression and a higher possibility of DR regression [117]. In addition, the DIRECT study showed a 34% DR regression in type II diabetic patients after 5 years of oral administration with Candesartan [118]. Hence, the use of RAS blockers should be considered at least to treat the early stages of DR, especially if RAS blockade is indicated.

### 3.4. Neurotrophic Factors

Neurotrophic factors are a family of molecules, mostly proteins, which are necessary for proper neuronal growth, differentiation and NVU interactions. Their deficiency during DR has been postulated as one of the main triggers of the neurodegenerative process that occurs in the diabetic retina [75,119]. Regarding the therapeutic implications that this group of molecules offers against DR, the most studied neurotrophic factors are pigment epithelial derived factor (PEDF), nerve growth factor (NGF), brain derived neurotrophic factor (BDNF), somatostatin (SST) and glucagon-like peptide 1 (GLP-1). The retinal levels of all of these have been found to be downregulated in diabetic retinas [120,121,122,123,124]. Furthermore, experimental therapies aimed at restoring the levels and functionality of these neuroprotective peptides have already been evaluated.

Intravenous and topical PEDF were both able to reduce inflammation, oxidative stress, glial activation and vascular leakage in streptozotocin-induced diabetic rats and Ins2Akita mice, respectively. The improvement in intravenous PEDF was accompanied by functional improvement [125,126]. Topical treatment with NGF in Ins2Akita mice prevented neurodegeneration and vascular abnormalities [127], while it reduced inflammation and apoptosis in diabetic rats [128]. BDNF delivered through adeno-associated virus improved the survival of retinal ganglion cells in streptozotocin-induced rats [129]; nonetheless, when high concentrations of BDNF are applied it worsens the inflammatory response [78]. SST eye drops prevented microglial activation, reactive gliosis and functional impairment in the retinas of db/db mice, while, in Bv.2 microglial cells, SST prevented the pro-inflammatory response [120]. In addition, in the EUROCONDOR clinical trial, topical administration of SST was found to be useful in preventing the worsening of preexisting retinal neurodysfunction in patients with type 2 diabetes aged 45 to 75 years with diabetes duration ≥5 years and ETDRS ≤35 [130]. Finally, topical administration of both GLP-1 receptor agonists and dipeptidyl peptidase-4 inhibitors exerted beneficial effects on the retinas of db/db mice, preventing oxidative stress [131,132], apoptosis, gliosis, inflammation, vascular leakage and functional abnormalities [133,134].

Nevertheless, new strategies based on increasing the bioavailability of these neurotrophic factors are needed (analogs, receptor agonists, inhibition of degradative enzymes, etc.) [135].

### 3.5. Stem Cell-Based Treatments

In recent years, there has been increasing interest in the use of stem-cell-based approaches for new therapies against retinal diseases [136]. Mesenchymal stem cells (MSCs) have multiple advantages, such as the absence of ethical issues, ease of harvesting, low risk of immunogenicity and tumorigenesis, and high capacity to reach injured areas. Regarding DR, MSCs have shown promising results in several in vitro and animal models, including against inflammation. In addition, genetic engineering of MSCs has also been successfully carried out to both overexpress retinal protective growth factors/cytokines and to optimize their arrival of MSCs to the site of injury. Finally, these beneficial effects have also been observed in clinical trials, despite the fact that some adverse effects have also been reported [137].

## 4. Conclusions

This review summarizes the latest findings regarding the role of the chronic inflammatory process that occurs in the diabetic retina as a consequence of the hyperglycemic conditions to which this tissue is exposed. Müller cells, astrocytes, microglia, leukocytes and the consequent release of pro-inflammatory cytokines, chemokines and adhesion molecules are the main players in this process, which contribute to all the hallmarks of the disease in both early and advanced stages. The purely anti-inflammatory treatments that are currently administered, such as glucocorticoids, NSAIDs or TNF-α blockers, exhibit partial beneficial effects and are associated with secondary adverse effects. Nonetheless, the latest research is providing new safer routes of administration, earlier interventions and new multitarget therapeutic strategies that confer protection not only against the inflammatory process that occurs during DR, but also to the other components of the disease, such as neurodegeneration or vascular alterations.

## Figures and Tables

**Figure 1 ijms-24-08796-f001:**
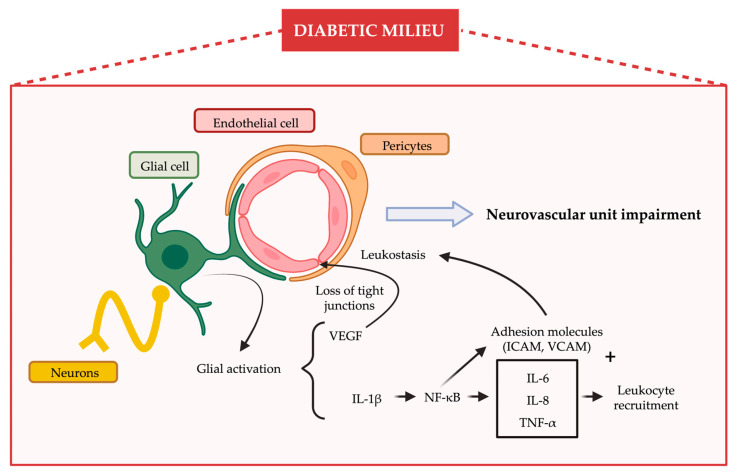
Inflammation and neurovascular unit impairment. Under hyperglycemic conditions, activated glial cells (reactive gliosis) produce several pro-inflammatory cytokines and vascular endothelial growth factor (VEGF) that increase vascular permeability. Moreover, a chemoattractant effect together with the expression of adhesion molecules promote leukocyte recruitment and leukostasis. These are some of the characteristic features that are observed at the initial neurovascular unit impairment.

**Figure 2 ijms-24-08796-f002:**
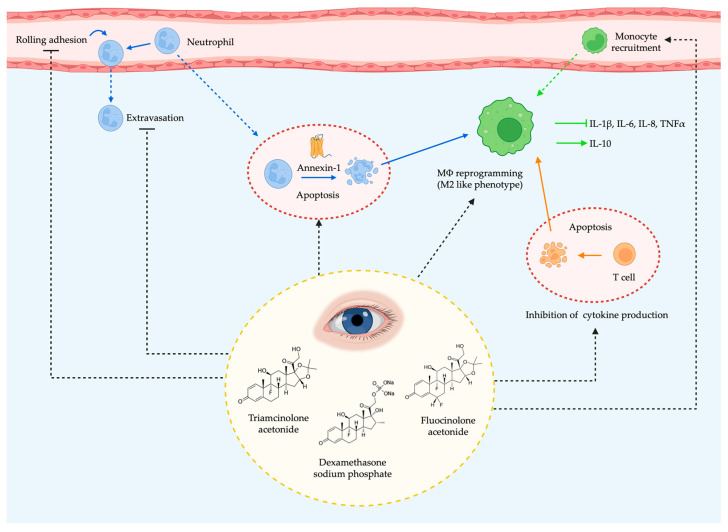
Main anti-inflammatory effects of corticosteroids in the diabetic retina. Glucocorticoids inhibit neutrophil adhesion and extravasation, promote neutrophil apoptosis by increasing Annexin-1 levels, promote T cell apoptosis, enhance monocyte recruitment and induce tissue resident macrophages (MΦ) to undergo a phenotypic change to become M2-like or anti-inflammatory. This new phenotype exhibits increased phagocytic activity to eliminate apoptotic cells and is no longer able to produce proinflammatory cytokines. Instead, these macrophages produce the anti-inflammatory cytokine IL-10.

## Data Availability

Not applicable.

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
