# Peer review of "Inflammation: The Link between Neural and Vascular Impairment in the Diabetic Retina and Therapeutic Implications"

_ijms, 2023, doi:10.3390/ijms24108796_

Round 1
Reviewer 1 Report
I carefully read the manuscript “Inflammation: the link between neural and vascular impairment in the diabetic retina. Therapeutic implications” by Ramos et al.
The Authors describe neural and vascular alterations occurring in diabetic retinopathy, focusing on the latest investigations on the role of inflammation processes. In particular they discuss about efficacy of currently administered anti-inflammatory treatments and those still under development.
In my opinion, the manuscript is well structured and the topic is interesting in the field of this ocular disease. I only suggest to include in the manuscript a further paragraph (or a few lines) mentioning recent developments using stem cell-based approaches. Some examples can be found in the following papers:
DOI: 10.4252/wjsc.v13.i6.632
DOI: 10.3390/ijms22094604
English Language is quite appropriate.
Author Response
Many thanks for the revision process and the kind comments on our manuscript. As recommended, a brief paragraph mentioning the recent development using stem cell-based approaches has been added to the revised manuscript (Section 3.5. Stem cell-based treatments)
Reviewer 2 Report
In this narrative review the authors present the latest literature evidence on
the link between neural and vascular diabetic retinopathy. The authors proposed the anatomical coupling of the different cellular components of the retina, in specific, neurons, glial cells, microglia, endothelial cells and pericytes,
affected by the glucose-rich vascular flow. The mechanism responsible for
the alterations of this complex tissue is the inflammation. The authors focused
in the the cellular and molecular components affected by the glucose high-triggered inflammation mediated by leukocytes, adhesion molecules and cytokines that underlie diabetic retinopathy. Therefore targeting inflammation in this disease is a promising therapeutic target for its treatment. In the last section of their review, the authors presented the latest information about anti-inflammatory drugs that can be used in the treatment of diabetic retinopathy.
Author Response
Thank you for your comprehensive comments on our paper. No further comments to be added from our side.
Reviewer 3 Report
Please find enclosed

Author Response
The manuscript is a well organized summary on inflammatory reaction in diabetic retina and
theraputic implations. It could be useful not only for scientists, but also for clinicians.
ANSWER: Many thanks for the revision process and the kind comments on our manuscript.
Line SDFI, additional information
Djuric Z et al. found that the SDF1-3’(801)AA genotype is more frequent in patients with proliferative diabetic retinopathy and pointed to a possible role of this allelic variant in the development of proliferative diabetic retinopathy. [Djuric Z, Sharei V, Rudofsky G, Morcos M, Li H, Hammes HP, et al. Association of homozygous SDF-13’A genotype with proliferative diabetic retinopathy. Acta Diabetol. 2010; 47: 79–82. doi: 10.1007/s00592-009-0119-2 PMID: 1938143230]
ANSWER: This information has been added at the end of the 3rd paragraph of the section 2.2. of the revised manuscript (“Inflammatory cytokines and their role in DR development”).
L318-346
For intravitreal use of steroids, a summary of drcrnet studies is inevitable and highly recommended
Kenalog: It must be mentioned, the intravitreal injection of Kenalog is off label. The summary of
product characteristics clearly indicates that the manufacturers do not recommend administration by intravitreal injection.
ANSWER: Some lines summarizing the results obtained in the DRCR.net studies regarding the intravitreal use of steroids have been added to section 3.1.1 of the revised manuscript (“ Corticosteroids”). In addition, we have now mentioned that intravitreal use of Kenalog is off-label.
L347-362
In everyday clinical practice, NSAIDs are mostly used to prevent and treat macular oedema after
cataract surgery. This problem is a serious one since the risk of macular edema after cataract surgery is significantly higher in diabetics than in non-diabetics.
Colin J. Chu et al: Risk Factors and Incidence of Macular Edema after Cataract Surgery A Database
Study of 81 984 Eyes. Ophthalmology 2016;123:316-323 ª 2016 by the American Academy of
Ophthalmology. This is an open access article under the CC BY-NC-ND license http://creativecommons.org/licenses/by-nc-nd/4.0/).
Singh RP et al: Efficacy of nepafenac ophthalmic suspension 0.1% in improving clinical outcomes
following cataract surgery in patients with diabetes: an analysis of two randomized studies. Clinical Ophthalmology 2017:11 1021–1029
Singh RP et al: Nepafenac 0.3% after Cataract Surgery in Patients with Diabetic Retinopathy Results of 2 Randomized Phase 3 Studies. Ophthalmology 2017;124:776-785 ª 2017 by the American Academy of Ophthalmology. This is an open access article under the CC BY-NC-ND license (http://creativecommons.org/licenses/by-nc-nd/4.0/).
ANSWER: A brief paragraph of NSAIDs role to treat macular edema after cataract surgery has been added using all the provided references (Section 3.1.2. Nonsteroidal anti-inflammatory drugs).
Reviewer 4 Report
Following the analysis of the manuscript titled "Inflammation: the link between neural and vascular impairment in the diabetic retina. Therapeutic implications", I appreciate the article's topic is interesting, the presentation of the information is clear and properly structured, and the figures are expressive and reinforce the presented notions.
I recommend that it should be revised taking into account the following observations:
- The manuscript is built too much on the presentation of own research, more studies should be consulted outside of these.
- Some references should be replaced as they are older than 10 - 15 years (37, 41, 43, 52, 55, 56, 62, 100, 112).
- Information about inflammation from lines 47-61 should be based on more than two references.
- Please insert an abbreviations list at the end of the manuscript.
- I suggest the introduction of a final section for the presentation of substances in clinical trials and the most promising results.
Author Response
Thank you very much for the revision process and the kind comments on our article. We have added more references apart from our own research and the references supporting information contained in lines 47-61 has been extended. In addition, references 37, 41, 43, 62, 112 have been replaced or removed. References 52,55, 56 and 100 have not been replaced because they belong to specific studies that, with the permission of the reviewer, we would like to continue to mention in this review article. As recommended a list of abbreviations has been added to the revised manuscript. Regarding the reviewer’s suggestion of adding a section on drugs in clinical trials indicating the most promising results, we feel that could be quite speculative and a source of potential bias and, therefore, we prefer to maintain the original version of the manuscript on this issue.
Round 2
Reviewer 3 Report
The authors addressed all issues in an appropriate manner.
Reviewer 4 Report
The new version of the manuscript is improved and can be published in its current form.